# Training intensity and improvements in exercise capacity in elderly patients undergoing European cardiac rehabilitation – the EU-CaRE multicenter cohort study

Thimo Marcin[1,2]*, Prisca Eser[1], Eva Prescott[3], Leonie F. Prins[4], Evelien Kolkman[4], Wendy Bruins[5], Astrid E. van der Velde[5], Carlos Peña Gil[6], Marie-Christine Iliou[7], Diego Ardissino[8], Uwe Zeymer[9], Esther P. Meindersma[5,10], Arnoud W. J. Van't Hof[5,11,12], Ed P. de Kluiver[5], Matthias Wilhelm[1]

1 Centre for Preventive Cardiology, Sports Medicine, Department of Cardiology, Bern University Hospital, University of Bern, Bern, Switzerland, 2 Graduate School for Health Sciences, University of Bern, Bern, Switzerland, 3 Department of Cardiology, Bispebjerg Frederiksberg University Hospital, Copenhagen, Denmark, 4 Diagram B.V., Zwolle, The Netherlands, 5 Isala Heart Centre, Zwolle, The Netherlands, 6 Department of Cardiology, Complexo Hospitalario Universitario de Santiago de Compostela, SERGAS IDIS CIBERCV, Santiago de Compostela, Spain, 7 Department of Cardiac Rehabilitation, Assistance Publique Hopitaux de Paris, Paris, France, 8 Department of Cardiology, Parma University Hospital, Parma, Italy, 9 Klinikum Ludwigshafen und Institut für Herzinfarktforschung Ludwigshafen, Ludwigshafen, Germany, 10 Department of Cardiology, Radboud University, Nijmegen, The Netherlands, 11 Department of Cardiology, Maastricht University Medical Center, and Cardiovascular Research Institute Maastricht (CARIM), Maastricht, The Netherlands, 12 Department of Cardiology, Zuyderland Medical Center, Heerlen, The Netherlands

* thimo.marcin@insel.ch

**Data Availability Statement:** Data cannot be shared publicly because of legislative restrictions in some of the participating countries. Data are

## Abstract

### Objectives

Guidelines for exercise intensity prescription in Cardiac Rehabilitation (CR) are inconsistent and have recently been discussed controversially. We aimed (1) to compare training intensities between European CR centres and (2) to assess associations between training intensity and improvement in peak oxygen consumption ($\dot{V}O_2$) in elderly CR patients.

### Methods

Peak $\dot{V}O_2$, heart rate and work rate (WR) at the first and second ventilatory thresholds were measured at start of CR. Training heart rate was measured during three sessions spread over the CR. Multivariate models were used to compare training characteristics between centres and to assess the effect of training intensity on change in peak $\dot{V}O_2$.

### Results

Training intensity was measured in 1011 out of 1633 EU-CaRE patients in 7 of 8 centers and the first and secondary ventilatory threshold were identified in 1166 and 817 patients, respectively. The first and second ventilatory threshold were found at 44% (SD 16%) and 78% (SD 9%) of peak WR and 78% (SD 9%) and 89% (SD 5%) of peak heart rate,

available upon request from the EU-CaRE steering comittee (contact via eu-care@diagram-zwolle.nl) for researchers who meet the criteria for access to confidential data.

**Funding:** For the Swiss consortium partner (TM, PE, MW), funding was received by the Swiss State Secretariat for Education, Research and Innovation under contract number 15.0139. All other authors received funding by the European Union's Horizon 2020 research and innovation programme under grant agreement No 634439. LP and EK received support in the form of a salary from Diagram B.V., a contract research organization. The specific roles of these authors are articulated in the 'author contributions' section. The funders had no role in study design, data collection and analysis, decision to publish, or preparation of the manuscript.

**Competing interests:** The authors have read the journal's policy and the authors of this manuscript have the following competing interests: AWJVH reports grants from Medtronic, grants and personal fees from Astra Zeneca, outside the submitted work; UZ reports grants and personal fees from Astra Zeneca, grants and personal fees from Bayer, personal fees from Boehringer Ingelheim, grants and personal fees from BMS, personal fees from Daiichi Sankyo, personal fees from Eli Lilly, grants and personal fees from Novartis, grants and personal fees from MSD, personal fees from Trommsdorf, personal fees from Amgen, outside the submitted work; LP and EK are paid employees of Diagram B.V. This does not alter our adherence to PLOS ONE policies on sharing data and materials. There are no patents, products in development or marketed products associated with this research to declare. All other authors have no Conflict of Interest to declare.

respectively. Training intensity and session duration varied significantly between centres but change in peak $\dot{V}O_2$ over CR did not. Training above the first individual threshold (β 0.62, 95% confidence interval [0.25–1.02]) and increase in training volume per hour (β 0.06, 95%CI [0.01–0.12]) were associated with a higher change in peak $\dot{V}O_2$.

## Conclusion

While training intensity and volume varied greatly amongst current European CR programs, changes in peak $\dot{V}O_2$ were similar and the effect of training characteristics on these changes were small.

## Introduction

Structured exercise training serving the purpose to improve exercise capacity and prognosis [1, 2] is a cornerstone of current comprehensive cardiac rehabilitation (CR). However, quantification of frequency, duration and especially intensity of exercise training varies between national and international CR guidelines [3].

The gold standard to prescribe exercise intensity is the determination of individual training domains (light-moderate, moderate-high, high-severe) defined by the first and secondary ventilatory thresholds ($VT_1$, $VT_2$) derived from cardiopulmonary exercise testing (CPET) [4]. However, these physiological thresholds are not readily detectable in all patients, and the determination thereof requires the conductance of CPET, which is not always available or feasible. Current guidelines suggest intensity prescription in percent of peak effort, ranging from 40% to 80% of peak oxygen consumption ($\dot{V}O_2$), 50–90% of peak heart rate (HR), or 40–70% of HR reserve [3]. Significant inconsistencies between different guidelines and discrepancies in threshold-based intensities were found in a recent study on patients undergoing CR [5], upon which the need of reconsidering the current guidelines for exercise prescription in the CR setting has been discussed [6]. The need for clearer guidelines, however, may only be indicated if training intensity plays an important role for the improvement in exercise capacity. A recent meta-analysis found significantly greater, though not clinically meaningful, improvements in peak $\dot{V}O_2$ with vigorous exercise interventions compared to interventions with lower intensities in a general CR population [7]. Despite the fact that guidelines recommend exercise above the $VT_1$, low intensities may also have a beneficial effect on exercise capacity, especially in cardiac patients with a significantly reduced pre-training exercise capacity [4] and patients with chronic heart failure [8]. The importance of training intensity in elderly cardiac patients has not been investigated thoroughly so far.

The aim of this study was (1) to compare training intensity domains derived from ventilatory thresholds with relative intensities of current guidelines in a large population of elderly cardiac patients and (2) to compare the training intensities utilized in different European CR centers and its influence on changes in peak $\dot{V}O_2$.

## Methods

The EU-CaRE study was a prospective cohort study, that assessed the (cost) effectiveness, sustainability and participation levels in current CR programs of eight cardiac rehabilitation centres in seven European countries (Denmark, France, Germany, the Netherlands, Italy, Spain and Switzerland). The study was approved by all relevant medical ethics committees:

Landesärztekammer Rheinland Pfalz, Germany (Nr. 837.341.15, (10109)); Comission Nationale de l'Informatique et de Libertés, France (DR-2016-021); medisch-ethische toetsingscommissie METC Isala Zwolle, The Netherlands (15.0350); Secretario do Comité de Ética da Investigación de Santiago-Lugo, Spain (2015/486); Comitato Ethico per Parma, Italy (34360); Videnskabsetiske Komite C for Region Hovedstaden, Denmark (593); Kantonale Ethikkomission Bern, Switzerland (290/15). The study was registered at trialregister.nl (NL5166). All participants gave written informed consent before they were included in the study.

## Study population

CR patients with an age of $\geq 65$ after an acute coronary syndrome, percutaneous intervention (PCI), CABG, surgical or percutaneous heart valve replacement (HVR) or documented coronary artery disease (CAD) were consecutively included from January 2016 –January 2018.

Patients with a contraindication to CR [9], mental impairment leading to inability to cooperate, severe impaired ability to exercise, signs of severe cardiac ischemia and/or a positive exercise testing on severe cardiac ischemia, insufficient knowledge of the native language and an implanted cardiac device were excluded.

## Data collection and processing

CPETs were performed on a cycle ergometer before and after CR using an individualised ramp protocol to achieve patient's voluntary exhaustion within 8 to 12 min of ramp duration. CPET raw data was processed in the core laboratory (Uni Bern) using MATLAB software (R2017, The MathWorks®, United States). To reduce a potential systematic bias for centres, all ventilatory thresholds (VT$_1$ and VT$_2$) were visually determined by one single investigator (TM), a sports scientist with extensive experience in setting ventilatory thresholds in healthy people as well as cardiac patients. The investigator was blinded to patient characteristics and centre. Interrater reliability was determined in a random subset of 200 CPETs, in which thresholds were determined by a second experienced investigator (MW) blinded also to patients and centres as well as to thresholds set by the other investigator. VT$_1$ was set at the beginning of a continuing increase of the ventilatory equivalent for oxygen ($\dot{V}_E/\dot{V}O_2$) without an increase in the ventilatory equivalent for carbon dioxide ($\dot{V}E/\dot{V}CO_2$) or beginning of a continuing increase in the end-tidal pressure of oxygen ($P_{ET}O_2$) without a decrease in the end-tidal pressure of carbon dioxide ($P_{ET}CO_2$), whichever was more discernible. VT$_2$ was set when there was a steeper $\dot{V}_E/\dot{V}CO_2CO_2$ increase or $P_{ET}CO_2$ decrease due to the exercise-induced lactic acidosis [10]. A centred moving average over 30 s for $\dot{V}O_2$, HR and WR was recorded at VT$_1$, VT$_2$ and peak exercise.

In each patient, we aimed to record HR during three training sessions, namely in a session during the first third of CR, during the middle of CR and towards the end of CR. The mean training HRs of a patient's monitored training sessions were averaged to one single mean training HR. In Copenhagen, Paris and Zwolle, training heart rate was measured with a mobile device and chest strap from MobiHealth B.V (Zwolle, The Netherlands). Ludwigshafen and Bern used stationary 3 channel electrocardiogram systems (Schiller Medizintechnik GmbH, München, Germany and ergoline GmbH, Bitz, Germany). Raw data of all monitored trainings except those from Parma and Nijmegen were processed in Bern using a MATLAB algorithm to smooth the HR signal and to filter noisy signals by robust local regression. Due to technical limitations, Parma provided HR (measured with ApexPro FH Telemetry system, GEHealthcare, U.S.) and training duration already averaged for each training and there were no monitored training sessions available for the small group of patients from Nijmegen (32 patients).

## Statistical analysis

All statistics were performed with R (Version 3.5.1, R Core Team, 2017).

Descriptive statistics included mean and standard deviation (SD) of $\dot{V}O_2$, HR, HR reserve and WR in percent of peak values at $VT_1$ and $VT_2$. Threshold values were given for a subgroup of patients reaching formal exertion during CPET at start of CR, defined by peak respiratory exchange ratio (RER = $\dot{V}CO_2/\dot{V}O_2$) >1.1. Level of agreement in VT setting within the random subset of 200 tests were assessed by Bland Altman plots. Training characteristics for each centre were reported by median and interquartile ranges for intensity in percent of HR peak and HR reserve at baseline CPET, average duration per session, total volume (duration × number of performed endurance sessions) and weekly volume. Proportion of patients with mean training intensity below their individual $VT_1$ was calculated for each centre.

Centre differences in training HR, duration, training volume and change in peak $\dot{V}O_2$ were tested using robust multivariate linear models (*robustbase* package) with centre as fixed factor and adjusted for the following potential confounders: age, sex, index intervention (PCI, CABG, surgical HVR, percutaneous HVR, documented CAD), HR at $VT_1$, beta-blocker, diabetes mellitus, days between index intervention and baseline CPET, and time between baseline CPET and recorded training in days. The effect of training intensity domain (training HR below vs. above individual $VT_1$) on change in peak $\dot{V}O_2$ was assessed by group comparison using Wilcoxon-rank sum test and with a multivariate linear mixed model (*lmer* package) with centre as random factor and additionally adjusted for the following fixed factors: age, sex, duration of CR, training volume per CR [h], peak $\dot{V}O_2$ at start of CR, index intervention and beta-blocker. Diagnostic plots were used to assess model assumptions. Alpha level was set at 0.05 for all analyses (two-tailed for Wilcoxon-rank sum test).

# Results

Overall, 1633 patients (mean age 72±5.4, 77% male) were included in the EU-CaRE study. Baseline characteristics were reported in detail elsewhere [11]. Fig 1 shows the flow chart of the available number of measured training intensities, ventilatory thresholds derived from the CPET at start of CR and outcome measures in change in peak $\dot{V}O_2$. Level of agreement of the ventilatory thresholds assessed by two investigators in a subset of CPETs is shown in S1 Fig and considered as acceptable.

Ventilatory thresholds reported in percent of different measures of peak effort are given in Table 1.

There were no large differences in thresholds relative to peak effort found between CPET at start and end of CR. The ventilatory thresholds relative to peak effort were slightly lower in the subgroup of patients who reached full exertion (RER ≥1.1). Fig 2 illustrates the training intensities measured at the first third, middle and last third of the CR duration as well as the ventilatory thresholds by centre.

In most centres, training intensity increased over the course of CR and in all centres except one, the majority of patients trained at an intensity between $VT_1$ and $VT_2$.(Table 2).

We found no significant centre differences with regard to change in peak $\dot{V}O_2$, as reported previously [12], despite significant differences in training intensity as well as training volume. Only one centre differed significantly from the average change in peak $\dot{V}O_2$, having also the lowest total training volume compared to all other centres.

Overall, from the subset of 808 patients in whom the training intensity domain could be determined, 519 (64%) exercised with an intensity above their individual $VT_1$ and improved

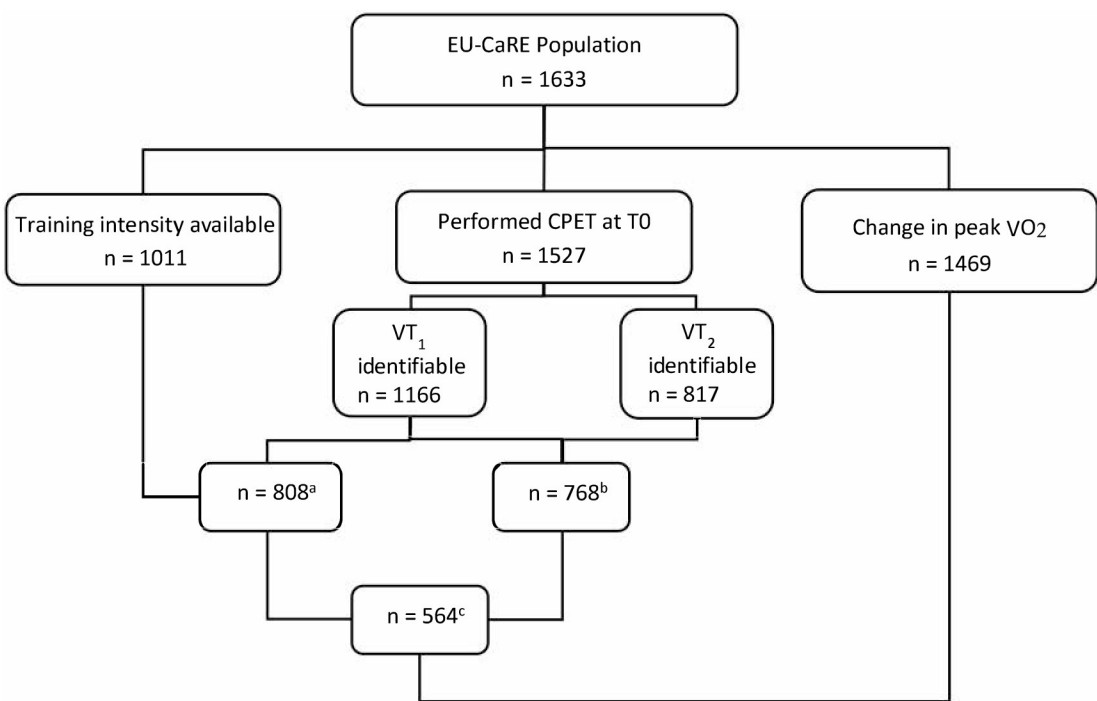

**Fig 1. Flow-chart of available cases.** N's represent number of patients having data on training intensity and/or ventilatory thresholds and/or change in peak $\dot{V}O_2$. [a] Number of patients with available training intensity and identified $VT_1$; [b]Number of patients with available training intensity and identified $VT_2$; [c] Number of patients with available training intensity and identified $VT_1$ and $VT_2$. CPET, cardiopulmonary exercise testing; $\dot{V}O_2$, oxygen consumption; T0, CPET at CR start; VT, ventilatory threshold.

their peak $\dot{V}O_2$ significantly more (+2.26 ml/kg/min in average) than patients who exercised with an intensity below the $VT_1$ (+1.63 ml/kg/min, Table 3).

In the multivariate mixed model, training above the individual $VT_1$ remained significantly associated with a higher improvement in peak $\dot{V}O_2$ [ml/kg/min] (β 0.62, 95% confidence interval 0.25–1.02). In addition, total training volume in hours per CR (β 0.06, 95%CI 0.01–0.12) was associated with a higher change in peak $\dot{V}O_2$. The interaction between intensity and volume was not significant and therefore removed from the model. The full output of the multivariate model is shown in S1 Table.

**Table 1. First and second individual ventilatory thresholds at start and end of CR relative to peak exercise.**

| | CPET start of CR all | | CPET end of CR all | | CPET start of CR subset RER≥1.1 | |
|---|---|---|---|---|---|---|
| | $VT_1$ | $VT_2$ | $VT_1$ | $VT_2$ | $VT_1$ | $VT_2$ |
| | (n = 1166) | (n = 817) | (n = 1280) | (n = 893) | (n = 546) | (n = 490) |
| %$\dot{V}O_2$ peak | 63 ± 11 | 84 ± 7 | 64 ± 12 | 88 ± 8 | 59±10 | 83±8 |
| %WR peak | 44 ± 16 | 78 ± 9 | 50 ± 14 | 85 ± 6 | 43±14 | 77±8 |
| %HR peak | 78 ± 9 | 89 ± 7 | 77 ± 9 | 91 ± 5 | 75±9 | 89±6 |
| %HR reserve | 45 ± 37 | 75 ± 22 | 50 ± 54 | 82 ± 40 | 42±19 | 74±19 |

CPET, cardiopulmonary exercise testing; CR, cardiac rehabilitation; VT, ventilatory threshold; $\dot{V}O_2$, oxygen consumption; HR, heart rate; WR, workrate; RER, respiratory exchange ratio

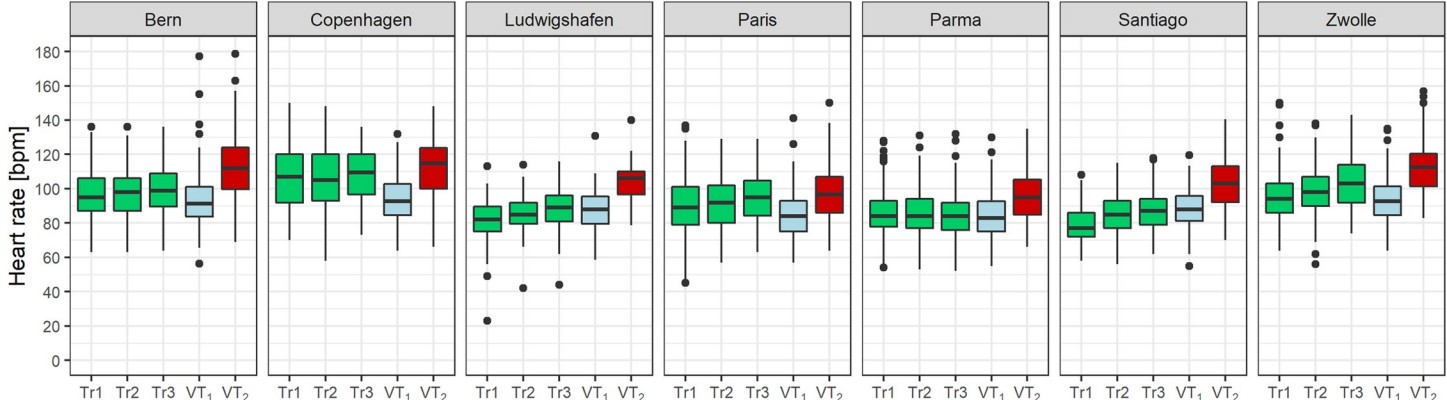

**Fig 2. Training heart rate (green) at beginning (Tr1), middle (Tr2) and end (Tr3) of CR compared to heart rate at first and second ventilatory threshold (VT$_1$, VT$_2$) at CR start for each centre.**

The mean (SD) change in peak $\dot{V}O_2$ in this subgroup of 808 patients included in the multivariate model was 2.04 (SD 2.74) ml/kg/min. The model explained 15.6% of the variance of change in peak $\dot{V}O_2$, with training intensity and training volume adding 0.8% and 0.5%, respectively, to the explained variance.

## Discussion

The present study provides data from a large cohort of elderly cardiac patients to compare current guidelines on exercise intensity with intensities derived from individual ventilatory threshold and compares training characteristics between seven European CR programs. Training characteristics varied widely between centres with total training volume ranging from 4.2 h to 19.3 h, training intensity from 73% to 85% of peak heart rate, and number of weeks from 2 to 21. In contrast, improvements in peak $\dot{V}O_2$ from start to end of CR varied little between centres. While training above the individual VT$_1$ and higher training volume were significantly associated with greater improvement in peak $\dot{V}O_2$, both variables explained less than one percent of the variance of the change in peak $\dot{V}O_2$.

### Ventilatory thresholds

According to the joint position statement of the European Association for Cardiovascular Prevention and Rehabilitation, the American Association of Cardiovascular and Pulmonary Rehabilitation and the Canadian Association of Cardiac Rehabilitation, the VT$_1$ is reached at around 50–60% of peak $\dot{V}O_2$ and 60–70% of peak HR whereas the VT$_2$ is reached at around 70–80% of peak $\dot{V}O_2$ and 80–90% of peak HR [4]. On average, the thresholds identified in this study are slightly higher than these ranges, probably due to a lesser degree of exertion reached in these elderly patients. However, the thresholds were only slightly lower relative to peak in a subgroup of patients who reached maximal exertion (RER≥1.1) and still relatively high when compared to the thresholds in the guidelines. This means that if training intensity was prescribed relative to peak $\dot{V}O_2$ or peak HR, the resulting training intensities tended to be below the target intensity. Our findings are in accordance to a previous study in patients with coronary artery disease, which found a large inter-individual variation, ranging from 47–91% of peak $\dot{V}O_2$ and 55–96% of peak HR [13]. Correspondingly, for 30% of our patients the target training HR was below the

**Table 2. Training characteristics and differences between centres.**

| *Training characteristics:* | Overall | Bern | Copenhagen | Ludwigshafen | Paris | Parma | Santiago | Zwolle |
|---|---|---|---|---|---|---|---|---|
| (frequencies) | | | | | | | | |
| Patients with measured training HR | 72% (1150 of 1601) | 81% (165 of 203) | 47% (111 of 237) | 24% (55 of 228) | 89% (196 of 219) | 90% (222 of 247) | 81% (199 of 247) | 92% (202 of 220) |
| Patients with identified $VT_1$ | 68% (1095 of 1601) | 84% (170 of 203) | 68% (162 of 237) | 50% (115 of 228) | 74% (161 of 219) | 56% (139 of 247) | 60% (149 of 247) | 90% (199 of 220) |
| Patients with training HR below $VT_1$ | 36% (304 of 848) | 29% (41 of 139) | 17% (15 of 88) | 46% (12 of 26) | 26% (38 of 144) | 42% (58 of 137) | 61% (79 of 130) | 33% (61 of 184) |
| (median and interquartile ranges) | | | | | | | | |
| Training intensity [% of HR peak[†]] | 80 [73; 87] | 78 [73; 84] | 83 [75; 89] | 81 [72; 85] | 84 [78; 91] | 85 [80; 90] | 73 [68; 79] | 79 [74; 86] |
| Training intensity [% of HR reserve[†]] | 52 [39; 65] | 54 [45; 62] | 64 [50; 76] | 47 [39;61] | 55 [42; 73] | 52 [33; 69] | 41 [32; 50] | 53 [46; 66] |
| Training Duration [min] | 29 [25; 35] | 33 [31; 35] | 30 [26; 34] | 22 [22; 23] | 29 [27; 31] | 20 [20; 27] | 54 [48; 59] | 26 [25; 27] |
| Total training volume [h][a] | 6.6 [4.6; 15.5] | 18.3 [15.5; 20.0] | 7.0 [5.8; 8.1] | 5.6 [5.1; 5.8] | 6.1 [5.3; 6.9] | 5.9 [4.4; 8.4] | 19.3 [16.8; 21.9] | 4.2 [3.8; 4.5] |
| Weekly training volume [min] | 61 [40; 124] | 55 [49; 68] | 30 [23; 34] | 120 [107; 133] | 126 [109; 139] | 182 [96; 215] | 56 [46; 68] | 36 [32; 40] |
| Training dose<br><br>(volume [h] * intensity [% of HR peak]) | 548 [381; 1236] | 1448 [1271; 1614] | 586 [470: 690] | 453 [392; 519] | 507 [443; 581] | 476 [375; 653] | 1404 [1263; 1593] | 330 [302; 361] |
| Change in peak $\dot{V}O_2$ [ml/kg/min] | 1.83 [0.35; 3.54] | 1.91 [-0.14; 4.66] | 1.63 [0.24; 3.65] | 2.03 [0.57; 3.53] | 2.82 [0.89; 4.34] | 1.69 [0.57; 3.17] | 1.52 [0.23; 2.93] | 1.50 [-0.20; 3.09] |
| *Centre differences* [b] *in:* | | | | | | | | |
| (ref: grand mean) | | | | | | | | |
| Training HR [bpm] | | 3.5** | 9.7*** | -6.8*** | 2.3* | -3.3** | -9.7*** | 4.4*** |
| Training Duration [min] | | 1.8** | -2.0** | -8.2*** | -2.1*** | -8.6*** | 24.7*** | -5.5*** |
| Total training volume [h] | | 8.8*** | -3.1*** | -3.3*** | -3.3*** | -3.2*** | 9.9*** | -5.7*** |
| Change in peak $\dot{V}O_2$ [ml/kg/min] | | 0.5 | 0.1 | 0.1 | 0.4 | -0.1 | -0.4 | -0.5* |

\* $p < 0.05$,

\*\* $< 0.01$,

\*\*\* $< 0.001$

† at CR start

[a] Total training volume is the mean duration of the endurance training session × attended number of endurance sessions over the course of CR

[b] Multivariate robust linear model adjusted for age, sex, index intervention, HR at VT1, Beta-blocker, Diabetes Mellitus, days between index intervention and start of CR, days between start of CR and training

$VT_1$. There is consensus in the CR community that threshold-based exercise intensity prescription is superior to intensities derived from peak values [4, 13, 14]. However, if CPETs cannot be performed or thresholds not identified, relative intensities are recommended [4]. This applied to approximately 25% of the patients in this study in whom $VT_1$ could not be identified, and approximately 50% of patients in whom $VT_2$ was not reached or could not be detected. On the other hand, around half of the patients did not reach an RER $\geq 1.1$ and therefore probably did not reach full exertion. However, a maximal or near-maximal effort is crucial for correct intensity prescription when using relative intensity domains [4]. Hence, prescription of optimal training intensities with current established methods may be difficult in elderly cardiac patients.

**Table 3. Patient characteristics according to training intensity domains.**

| | Training Intensity Domains | | | |
| --- | --- | --- | --- | --- |
| | Light-moderate (below $VT_1$) N = 289[1] | Moderate-high (above $VT_1$) N = 519[1] | p-value[2] | Missing[3] N = 825[1] |
| Age [years] | 72.7 (5.2) | 72.1 (5.0) | 0.17 | 73.4 (5.7) |
| Male sex | 231 (80%) | 436 (84%) | 0.17 | 592 (72%) |
| Index Intervention | | | 0.53 | |
|  CABG | 84 (29%) | 170 (33%) | | 227 (28%) |
|  Chronic CAD no intervention) | 22 (7.6%) | 34 (6.6%) | | 40 (4.8%) |
|  PCI | 157 (54%) | 274 (53%) | | 459 (56%) |
|  Percutaneous treated VHD | 2 (0.7%) | 8 (1.5%) | | 23 (2.8%) |
|  Surgical treated VHD | 24 (8.3%) | 33 (6.4%) | | 76 (9.2%) |
| Betablocker | 241 (83%) | 422 (81%) | 0.52 | 664 (80%) |
| CR duration [days] | 79 (57) | 71 (52) | 0.094 | 63 (52) |
| Total aerobic training hours per CR [hours] | 10.9 (7.4) | 9.2 (6.2) | **0.027** | 9.0 (6.1) |
| Peak $\dot{V}O_2$ at start of CR [ml/kg/min] | 16.9 (4.4) | 17.6 (4.7) | **0.025** | 14.5 (4.6) |
| Peak $\dot{V}O_2$ at end of CR [ml/kg/min[ | 18.5 (4.7) | 19.9 (5.2) | **<0.001** | 16.4 (4.8) |
| Change in peak $\dot{V}O_2$ [ml/kg/min] | 1.63 (2.48) | 2.26 (2.84) | **0.003** | 2.00 (2.80) |
| Change in peak $\dot{V}O_2$ [% of baseline] | 11 (16) | 14 (18) | **0.008** | 17 (26) |

[1] Statistics presented: mean (SD); n (%)

[2] Statistical tests performed for training intensity group differences: Wilcoxon rank-sum test; chi-square test of independence; Fisher's exact test

[3] Patients with missing data for training heart rate (n = 613), VT1 (n = 508) or change in peak $\dot{V}O_2$ (n = 51)

$VT_1$, first ventilatory threshold; CABG, coronary artery bypass grafting; CAD, coronary artery disease; PCI, percutaneous coronary intervention; VHD, valvular heart disease; CR, cardiac rehabilitation; $\dot{V}O_2$, oxygen consumption

## Training intensity

Training HR was between HR at $VT_1$ and $VT_2$ in the majority (64%) of patients and consequently in the range of the moderate to high-intensity domain [4]. Nevertheless, a considerable proportion of patients exercised at a HR below their individual $VT_1$, ranging from 17% to 61% in different centres, despite the widely endorsed recommendation for progression from moderate to vigorous intensity exercise over the duration of CR [3]. However, evidence exists that also low exercise intensity may be effective in cardiac patients with reduced exercise capacity [4]. In this study, changes in peak $\dot{V}O_2$ did not differ greatly between centres, despite differences in training volume and training intensity.

On a patient level on the other hand, a higher training intensity and greater training volume were significantly associated with increased changes in peak $\dot{V}O_2$ over the course of CR. Patients who trained at an intensity above their individual $VT_1$ increased their peak $\dot{V}O_2$ on average by 0.63 ml/kg/min more than patients who trained below their individual $VT_1$. This relation remained stable in the multivariate model adjusted for potentially confounding factors (such as centres) with a 0.62 ml/kg/min significantly greater change in peak $\dot{V}O_2$ in patients who exercised above $VT_1$. Whether this difference was clinically meaningful is questionable, while it corresponded to little more than a quarter of the mean change in peak $\dot{V}O_2$, both, a 14% and 11% improvement are relatively small. In comparison, a recent meta-analysis on 13'220 patients of 128 studies (mean age 58.4) found an additional improvement of 1.5 ml/kg/

min over the course of CR through prescription of higher intensities, which the authors did not consider as clinically relevant [7]. In addition, a higher total training volume achieved during CR was associated with a greater change in peak $\dot{V}O_2$, corresponding to 0.07 ml/kg/min increase in peak $\dot{V}O_2$ for every one-hour increase in total training volume during CR. Each metabolic equivalent (MET, 3.5 ml/kg/min of $\dot{V}O_2$) increase in exercise capacity during CR has previously been found to be associated with 13% lower mortality [15]. Accordingly, exercise training for 50 hours may increase peak $\dot{V}O_2$ by one MET. Patients exercising above the $VT_1$ may achieve one MET improvement with fewer training hours, although this relation is not supported by our data (no significant interaction between training volume and intensity).

In summary, our results suggest that even patients who exercise at an intensity below their individual $VT_1$ improved their peak $\dot{V}O_2$, although somewhat less than those exercising above. This suggests that the focus on specific training intensities may be overrated in elderly cardiac patients.

## Self-paced intensity instead of redefining prescription

Given the difficulties of determining ventilatory thresholds or using relative intensity domains in elderly patients, as well as the potentially low impact of training intensity on change in exercise capacity, a self-paced approach seems warranted in elderly cardiac patients.

Already widely established in clinical routine is the exercise intensity prescription according to self-rated perceived exertion using the BORG scale [4]. This method, while providing scope for patient autonomy, allows clinicians to direct patients towards different intensity ranges.

A recent meta-analysis found better affective response after self-selected exercise intensities [16]. However, the differences between self-paced and prescribed training intensities were mainly driven by studies that prescribed training intensity above the $VT_1$, while studies with training intensities below the $VT_1$ did not find differences with regard to affective response. Intensities above the $VT_1$ were found to evoke greater negative affective response than self-selected exercise performed at lower intensities [17]. However, cardiac patients, and in particular, elderly cardiac patients were underrepresented in these studies. Nevertheless, it seems plausible that elderly cardiac patients would prefer self-selected or lower intensities. Patients' preference for cardiac rehabilitation delivery has recently gained attention and home-based cardiac rehabilitation was discussed as valid alternative to centre-based CR [18, 19]. In view of the growing interest in personalised therapy, it would be more appropriate to direct the focus on patients' preferences instead of redefining exercise intensity prescriptions. The beneficial effect of exercise is likely to be abolished when patients discontinue exercising after CR. Larger studies are warranted to assess if self-paced training intensities are feasible, safe and equally (sustainably) effective to prescribed intensities in elderly cardiac rehabilitation patients.

## Limitations

We did not differentiate between training modalities, despite the fact that two centres (Copenhagen, Zwolle) performed high intensity interval trainings while the other centres mostly performed moderate continuous training. However, it seems unlikely that modality had a major impact on changes in exercise capacity as patients from Copenhagen and Zwolle did not differ largely from other centers with regard to changes in peak $\dot{V}O_2$. Also, we did not assess habitual physical activity outside of the CR program which may have influenced changes in peak $\dot{V}O_2$. Additionally, results of the present study do not reflect the whole population of the EU-CaRE study as only patients with monitored training sessions, and only those with good quality CPET (that allowed the determination of $VT_1$) could be included in the multivariate models.

Baseline peak $\dot{V}O_2$ of the patients included in the model was 17.36 ml/kg/min, while the mean baseline peak $\dot{V}O_2$ overall EU-CaRE patients was 15.94 ml/kg/min. Therefore, we do not know whether weaker patients could also increase their exercise capacity by one MET if they exercised for 50 hours. Nevertheless, this is, to the best of our knowledge, the first analysis relating accurately monitored exercise intensity to change in peak $\dot{V}O_2$ in such a large data set of elderly CR patients.

## Conclusion

Overall, training intensities of our elderly CR population followed current guidelines. While training intensities above the individual $VT_1$ were associated with greater improvement in peak $\dot{V}O_2$, the association was weak. Despite large differences in training intensities between current European CR programmes, improvements in exercise capacity were very similar. Therefore, the superiority of certain training prescription over others needs to be questioned and the focus on specific training intensities may be overrated in elderly patients.

In a quarter of our elderly CR cohort, the ventilatory thresholds could not be determined and full exertion (RER > 1.1) was not reached in about half of our cohort. Accurate prescription of exercise intensity may therefore often not be possible. Future studies on safety and efficacy of self-paced exercise intensity in elderly cardiac rehabilitation patients are warranted.

## Supporting information

**S1 Fig. Reproducibility of ventilatory threshold setting in a subset of 200 randomly chosen cardiopulmonary exercise tests.**
(DOCX)

**S1 Table. Linear mixed model on change in peak $\dot{V}O_2$ [ml/kg/min] with centre as random factor.**
(DOCX)

## Author Contributions

**Conceptualization:** Thimo Marcin, Prisca Eser, Matthias Wilhelm.

**Data curation:** Thimo Marcin, Leonie F. Prins, Evelien Kolkman.

**Formal analysis:** Thimo Marcin, Prisca Eser.

**Funding acquisition:** Eva Prescott, Wendy Bruins, Astrid E. van der Velde, Arnoud W. J. Van't Hof, Ed P. de Kluiver.

**Investigation:** Eva Prescott, Carlos Peña Gil, Marie-Christine Iliou, Diego Ardissino, Uwe Zeymer, Esther P. Meindersma, Arnoud W. J. Van't Hof, Ed P. de Kluiver, Matthias Wilhelm.

**Methodology:** Eva Prescott, Carlos Peña Gil, Marie-Christine Iliou, Diego Ardissino, Uwe Zeymer, Esther P. Meindersma, Arnoud W. J. Van't Hof, Ed P. de Kluiver, Matthias Wilhelm.

**Project administration:** Leonie F. Prins, Wendy Bruins, Astrid E. van der Velde.

**Resources:** Eva Prescott, Carlos Peña Gil, Marie-Christine Iliou, Diego Ardissino, Uwe Zeymer, Esther P. Meindersma, Ed P. de Kluiver, Matthias Wilhelm.

**Software:** Evelien Kolkman.

**Supervision:** Arnoud W. J. Van't Hof.

**Visualization:** Thimo Marcin.

**Writing – original draft:** Thimo Marcin, Prisca Eser.

**Writing – review & editing:** Eva Prescott, Leonie F. Prins, Evelien Kolkman, Wendy Bruins, Astrid E. van der Velde, Carlos Peña Gil, Marie-Christine Iliou, Diego Ardissino, Uwe Zeymer, Esther P. Meindersma, Arnoud W. J. Van't Hof, Ed P. de Kluiver, Matthias Wilhelm.

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
