## [Decision Letter · Decision Letter 0]

10 Sep 2020

PONE-D-20-24527

Training intensity and improvements in exercise capacity in elderly patients undergoing European cardiac rehabilitation – The EU-CaRE multicenter cohort study

PLOS ONE

Dear Dr. Marcin,

Thank you for submitting your manuscript to PLOS ONE. After careful consideration, we feel that it has merit but does not fully meet PLOS ONE’s publication criteria as it currently stands. Therefore, we invite you to submit a revised version of the manuscript that addresses the points raised during the review process.

We look forward to receiving your revised manuscript.

Kind regards,

Corstiaan den Uil

Academic Editor

PLOS ONE

Journal Requirements:

2. Thank you for stating the following in the Competing Interest section:

"AWJVH reports grants from Medtronic, grants and personal fees from Astra Zeneca, outside the submitted work, UZ reports grants and personal fees from Astra Zeneca, grants and personal fees from Bayer, personal fees from Boehringer Ingelheim, grants and personal fees from BMS, personal fees from Daiichi Sankyo, personal fees from Eli Lilly, grants and personal fees from Novartis, grants and personal fees from MSD, personal fees from Trommsdorf, personal fees from Amgen, outside the submitted work

All other authors have no Conflict of Interest to declare."

We note that one or more of the authors are employed by a commercial company: Diagram B.V., Zwolle.

2.1. Please provide an amended Funding Statement declaring this commercial affiliation, as well as a statement regarding the Role of Funders in your study. If the funding organization did not play a role in the study design, data collection and analysis, decision to publish, or preparation of the manuscript and only provided financial support in the form of authors' salaries and/or research materials, please review your statements relating to the author contributions, and ensure you have specifically and accurately indicated the role(s) that these authors had in your study. You can update author roles in the Author Contributions section of the online submission form.

2.2. Please also provide an updated Competing Interests Statement declaring this commercial affiliation along with any other relevant declarations relating to employment, consultancy, patents, products in development, or marketed products, etc.  

Reviewers' comments:

Reviewer's Responses to Questions

**Comments to the Author**

1. Is the manuscript technically sound, and do the data support the conclusions?

Reviewer #1: Yes

Reviewer #2: Partly

Reviewer #3: Yes

2. Has the statistical analysis been performed appropriately and rigorously? 

Reviewer #1: I Don't Know

Reviewer #2: Yes

Reviewer #3: Yes

3. Have the authors made all data underlying the findings in their manuscript fully available?

Reviewer #1: Yes

Reviewer #2: Yes

Reviewer #3: Yes

4. Is the manuscript presented in an intelligible fashion and written in standard English?

Reviewer #1: Yes

Reviewer #2: Yes

Reviewer #3: Yes

5. Review Comments to the Author

Reviewer #1: The subject of this study is very relevant at this moment: I therefor thank the authors for performing this analysis: the percentage of (elderly) patients attending and completing CR remains too low worldwide.

The following items I think is missing/unanswered:

Was there a difference in results between pts LVEF above vs below 40% or between ACS vs stable AP vs valvereplacement pts?

What was the percentage of patients who completed the whole CR program offered per center? Why was the ventilator thresholds determined by 1 single investigator only?

Do you think there was a selection bias because only

patients with monitored training sessions, and only those with good quality CPET (that

allowed the determination of VT1) could be included in the multivariate models? Did this greatly influenced your results?

You advocate " self-paced-intensity", especially because this is an elderly population: I completely agree: do you know what the relation was between the achieved training intensities and improvement in Quality of Life in your study group? What are the reasons that Copenhagen and Zwolle chose for high intensity interval trainings?

Reviewer #2: This is an interesting paper with findings that could be clinically highly relevant. However, the reviewer did notice some issues in the methodology and results that require further (statistical) analysis, or at least significant reconsideration, to come to valid conclusions. Therefore, to increase the likelihood for publication, a major revision may be needed.

Introduction

It could be explained in greater detail to what extend VO2peak matters in older patients with CVD (e.g. prognosis, physical functioning, quality of life)?

Please include a study hypothesis?

Methods

What is ‘old’? For now, the cut-off of 65 years was used (which is arbitrary, but I understand the authors would select this age threshold), but in CR practice, 65 years is often the average age of the population (see for example reference 3). So I wonder whether similar results will be found if the authors would specifically focus on patients >75 years?

‘In each patient, we aimed to record HR during three trainings…’ should be ‘In each patient, we aimed to record HR during three training sessions…’?

Were changes in beta-blocker therapy allowed during CR? This could affect the training HR and consequently the statistical outcomes.

‘Alpha level was set at 0.05 for all analysis’: was this two-tailed?

Were the patients allowed to execute strength training (older patients are often in need of strength training...)? If so, was this corrected for during the statistical analysis?

Results

Please mention n/centre in Table 2.

In Table 2, total training volume should be expressed as peak effort training hours as well (sessions * duration/session * intensity). For now, the actual volume of exercise (expressed as hours) cannot be compared between the centres if the intensity also varied (to be determined by univariate correlations). In addition, would these peak effort training hours correlate with changes in VO2peak and thus affect the outcome from the regression analysis?

For now, the authors compared patients training below or above VT1. I’m not surprised that patients exercising above VT1 improve better when compared with patients exercising below VT1 (below VT1 is really low…). However, between VT1 and VT2 a large HR range can be prevalent in some patients. So the question arises whether the effects of exercise intensity would be seen if patients would be categorized according to a different % of VT2 (under the assumption that the patients did not exercise above VT2)?

Discussion

Habitual physical activity was not assessed during CR: this could have affected the change in VO2peak and should be acknowledged as a limitation.

The authors mention: ‘According to the joint position statement of the European Association for Cardiovascular Prevention and Rehabilitation, the American Association of Cardiovascular and Pulmonary Rehabilitation and the Canadian Association of Cardiac Rehabilitation, the VT1 is reached at around 50-60% of peak VO2 and 60-70% of peak HR whereas the VT2 is reached at around 70-80% of peak VO2 and 80-90% of peak HR. [2] On average, the thresholds identified in this

study are slightly higher than these ranges, probably due to a lesser degree of exertion reached in these elderly patients.’ This actually is not an issue that should be considered as a study limitation: reference 3 reports similar thresholds…so they actually agree with literature. The reviewer believes rather the guidelines should change according to observations made in many CR centres.

Reviewer #3: This paper presents results from a prospective, multi-centre study investigating the cost effectiveness and sustainability of, and engagement in, cardiac rehabilitation (CR) programmes across several European countries. The analyses presented within this paper are a secondary analysis of the data, examining the intensity and volume of exercise completed during CR compared to current guidelines, and examine their association with improvements in directly measured V̇O2peak. A major strength of this study is the comparatively large sample size allowing for comprehensive multivariate modelling. The paper is novel in that it surveys exercise training variables within current CR programmes compared to international guidelines. Moreover, this paper is the first to examine the role of intensity and volume on cardiorespiratory fitness outcomes in a large cohort of older adults.

The paper is generally well written, though may benefit from minor edits as noted below.

General:

• Some inconsistencies in formatting of VO2 throughout.

• Incorrect formatting for abbreviations of physiological measures: VE requires subscript E, and VO2, VCO2 and VE all require an over-dot above the V to indicate values are per unit time (e.g. V̇O2 and V̇E).

• I strongly recommend referring to workload as work rate (abbreviated to WR in the results) as opposed to Watt – it is not common practice to refer to a variable by its units (e.g. the authors do not refer to heart rate as bpm)

Introduction:

• Formatting of list for aims should present numbers as (1) and (2).

Methods:

• The authors should elaborate on, or provide citation for, contraindications to cardiac rehabilitation as part of their exclusion criteria

• The methods state that thresholds were only analysed by a single author. The authors should provide a rationale for this and note the experience/expertise of the lead author to reliably identify ventilatory thresholds. Moreover, did the authors conduct any assessment of reliability for a random subset of CPET outputs? If not, I believe this should be undertaken and the agreement between authors presented within the results. As this is a major outcome for this study it is imperative the authors demonstrate the accuracy and reliability of their analyses.

• 30 s moving average for V̇O2, HR and Watts at VT1 and VT2 requires further clarification – was this taken as the 30 s prior or 30 s following, or 15 s either side?

• When determining whether a patient’s mean exercise intensity was above or below VT1, how was specifically determined? Did the authors calculate the mean HR for each session then compare the mean of these means to the HR at VT1? This requires further clarification.

Results:

• The authors should undertake secondary confirmation of thresholds identified using a second author. Given the large sample size this may be completed as a random subsample of 15-20% of participants. The authors should then report the level of agreement in identifying VT1 and VT2 in this subsample.

• Figure 1 is difficult to interpret. This could be made clearer by additional descriptions for the three lowest boxes describing what the n values represent (i.e. No. of pts with both VT1 and VT2, n = 768) [I am assuming that is what this box represents?]

• I believe a total cohort version of Figure 2 would be beneficial to include in the results. This could be included as an additional figure rather then adding further complexity to Figure 2.

• Legibility of Figure 2 could be improved by reducing line thickness of the boxplots and size of outlier symbols. Further, consider a different (lighter) colour scheme for the green and red boxes as these colours reduce the contrast with the box outline.

• There are abbreviation inconsistencies in caption for Figure 2.

• Can the authors include an additional column in Table 2 for the total cohort. This should be placed before the column for Bern. Naturally, present data for frequencies and median/IQR sections only.

• In Table 2, footnote markers for a and b should be swapped – currently b comes before a.

Discussion:

• Final statement in ‘Training intensity’ subsection could be elaborated on. The authors should clarify if they suggest that intensity is not an important consideration for exercise prescription in this population – if not, what should clinicians focus on?

• The self-paced exercise intensity subsection should be expanded to include discussion regarding perceptually-regulated exercise (i.e. RPE regulated). In practice, there is a continuum of self-regulated/self-paced exercise intensity. The authors have discussed affect-regulated exercise which encompasses exercising at an intensity that ‘feels good’ or elicits a positive affective response. In most circumstances, this is method of regulating exercise intensity introduces a high level of subjectivity; the clinician cannot effectively guide the patient to exercise at a low, moderate or high intensity. In contrast, with RPE-regulated exercise provides scope for patient subjectivity but still allows the clinician to direct the patient to different general intensities. This should be discussed in reference to the results of the present study that suggest that the intensity may not be important and that exercise at intensities that elicit a positive affective response may (only some emerging data is available) be associated with better attendance and compliance in CR programmes.

• The limitations section should not that thresholds used in analyses were identified by only 1 author. The authors should consider and discuss the impact of erroneous threshold values on their outcomes.

6. PLOS authors have the option to publish the peer review history of their article (what does this mean?). If published, this will include your full peer review and any attached files.

Reviewer #1: No

Reviewer #2: No

Reviewer #3: No

---

## [Author Response · Author response to Decision Letter 0]

30 Oct 2020

We would like to thank the reviewers for the very valuable comments, which we have gladly addressed and hopefully implemented to the editor’s and reviewers’ expectations. Many comments were very valid and we feel that the presented changes have greatly improved the clarity of the paper. Please find our point-to-point responses (red) in the uploaded file "response to the editor and reviewers".

---

## [Decision Letter · Decision Letter 1]

4 Nov 2020

Training intensity and improvements in exercise capacity in elderly patients undergoing European cardiac rehabilitation – The EU-CaRE multicenter cohort study

PONE-D-20-24527R1

Dear Dr. Marcin,

We’re pleased to inform you that your manuscript has been judged scientifically suitable for publication and will be formally accepted for publication once it meets all outstanding technical requirements.

Kind regards,

Corstiaan den Uil

Academic Editor

PLOS ONE

Additional Editor Comments (optional):

Reviewers' comments:

Reviewer's Responses to Questions

**Comments to the Author**

1. If the authors have adequately addressed your comments raised in a previous round of review and you feel that this manuscript is now acceptable for publication, you may indicate that here to bypass the “Comments to the Author” section, enter your conflict of interest statement in the “Confidential to Editor” section, and submit your "Accept" recommendation.

Reviewer #2: All comments have been addressed

Reviewer #3: All comments have been addressed

2. Is the manuscript technically sound, and do the data support the conclusions?

Reviewer #2: Yes

Reviewer #3: Yes

3. Has the statistical analysis been performed appropriately and rigorously? 

Reviewer #2: Yes

Reviewer #3: Yes

4. Have the authors made all data underlying the findings in their manuscript fully available?

Reviewer #2: Yes

Reviewer #3: No

5. Is the manuscript presented in an intelligible fashion and written in standard English?

Reviewer #2: Yes

Reviewer #3: Yes

6. Review Comments to the Author

Reviewer #2: This is now a good and clinically relevant manuscript that can be published in its current form. There are no further suggestions.

Reviewer #3: Thank you for the opportunity to review the manuscript. I am satisfied with the edits performed and responses to reviewer comments. I believe the manuscript to be a valuable contribution to the cardiac rehabilitation field and recommend publication in its current state. I have no further comments for the revised manuscript.

7. PLOS authors have the option to publish the peer review history of their article (what does this mean?). If published, this will include your full peer review and any attached files.

Reviewer #2: No

Reviewer #3: No

---

## [Editor Report · Acceptance letter]

6 Nov 2020

PONE-D-20-24527R1 

Training intensity and improvements in exercise capacity in elderly patients undergoing European cardiac rehabilitation – The EU-CaRE multicenter cohort study 

Dear Dr. Marcin:

I'm pleased to inform you that your manuscript has been deemed suitable for publication in PLOS ONE. Congratulations! Your manuscript is now with our production department. 

Kind regards, 

on behalf of

Dr. Corstiaan den Uil 

Academic Editor

PLOS ONE